# Changes in Sex Life among People in Taiwan during the COVID-19 Pandemic: The Roles of Risk Perception, General Anxiety, and Demographic Characteristics

**DOI:** 10.3390/ijerph17165822

**Published:** 2020-08-11

**Authors:** Nai-Ying Ko, Wei-Hsin Lu, Yi-Lung Chen, Dian-Jeng Li, Yu-Ping Chang, Chia-Fen Wu, Peng-Wei Wang, Cheng-Fang Yen

**Affiliations:** 1Department of Nursing, College of Medicine, National Cheng Kung University, Tainan 70101, Taiwan; nyko@mail.ncku.edu.tw; 2Department of Psychiatry, Ditmanson Medical Foundation Chia-Yi Christian Hospital, Chia-Yi City 60002, Taiwan; wiiseen@gmail.com; 3Department of Healthcare Administration, Asia University, Taichung 41354, Taiwan; elong@asia.edu.tw; 4Department of Psychology, Asia University, Taichung 41354, Taiwan; 5Department of Addiction Science, Kaohsiung Municipal Kai-Syuan Psychiatric Hospital, Kaohsiung 80276, Taiwan; u108800004@kmu.edu.tw; 6Department of Psychiatry, School of Medicine, and Graduate Institute of Medicine, College of Medicine, Kaohsiung Medical University, Kaohsiung 80708, Taiwan; pino3015@hotmail.com; 7School of Nursing, The State University of New York, University at Buffalo, NY, NY 14214-3079, USA; yc73@buffalo.edu; 8Department of Psychiatry, Kaohsiung Medical University Hospital, Kaohsiung 80708, Taiwan

**Keywords:** COVID-19, sex life, risk perception, anxiety, sexual minority

## Abstract

This study used data collected from an online survey study on coronavirus disease 2019 (COVID-19) in Taiwan to examine changes in sex life during the pandemic and the factors affecting such changes. In total, 1954 respondents were recruited from a Facebook advertisement. The survey inquired changes in sex life during the pandemic, including satisfaction with the individual’s sex life, frequency of sexual activity, frequency of sex-seeking activity, and frequency of using protection for sex. The associations of change in sex life with risk perception of COVID-19, general anxiety, gender, age, and sexual orientation were also examined. For each aspect of their sex life, 1.4%–13.5% of respondents reported a decrease in frequency or satisfaction, and 1.6%–2.9% reported an increase in frequency or satisfaction. Risk perception of COVID-19 was significantly and negatively associated with frequencies of sexual and sex-seeking activities. Higher general anxiety was significantly and negatively associated with satisfaction of sex life and frequencies of sexual and sex-seeking activities. Sexual minority respondents were more likely to report decreased satisfaction with sex life and frequencies of sexual activity and sex-seeking activities during COVID-19. Health care providers should consider these factors when developing strategies for sexual wellness amid respiratory infection epidemics.

## 1. Introduction

### 1.1. The Coronavirus Disease 2019 (COVID-19) Pandemic in Taiwan

COVID-19 emerged in Wuhan, China, at the end of 2019 and has been spread to over 200 countries and territories around the world, with the total number of infected cases having risen to over 14,000,000 and more than 590,000 deaths, as of 17 July 2020 [1]. Given the geographical proximity and busy transportation between Taiwan and China, Taiwan was predicted to have the second highest number of COVID-19 cases early in the outbreak [2]. After all, Taiwan experienced the severe outbreak of 2002–2003 Severe Acute Respiratory Syndrome (SARS), which also originated from China; globally, Taiwan had the third highest number of SARS cases, after China and Hong Kong [3]. The experience with SARS made the Taiwanese government and people vigilant against COVID-19. Early since December 31, 2019, Taiwanese officials began to assess passengers on direct flights from Wuhan for fever and pneumonia symptoms before passengers could deplane.

The first COVID-19 case in Taiwan was confirmed on 21 January 2020 [4]. During the period from January 20 to February 24, the Taiwan Centers for Disease Control rapidly produced and implemented a list of at least 124 action items including border control, case identification, quarantine of suspicious cases, proactive case finding, resource allocation, reassurance and education of the public while fighting misinformation, negotiation with other countries and regions, formulation of policies toward schools and childcare, and relief to businesses [5]. With proactive containment efforts and comprehensive contact tracing, the number of COVID-19 cases in Taiwan remained low, as compared with other countries that had widespread outbreaks [6]. Therefore, there was no social lockdown in Taiwan. As of 17 July 2020, Taiwan had tested a total of 79,395 persons showing 454 confirmed cases, of which only 55 were domestic. Six patients died, and 440 people were released from hospital after testing negative three times subsequently [1].

### 1.2. Sex Life during the COVID-19 Pandemic

The COVID-19 pandemic results in widespread and wide-ranging concerns spanning the issues of physical morbidity and mortality [2], mental health [7,8], the economy [9], education [10], and interpersonal relationships [11]. Sexual health in the context of the pandemic, however, warrants investigation [12]. According to the World Health Organization, sexual health is a state of physical, mental, and social well-being in relation to sexuality [13]; sexuality is influenced by the interaction of biological, psychological, social, economic, political, cultural, legal, historical, religious, and spiritual factors [13]. Accordingly, it is reasonable to hypothesize that sex life would be deeply influenced by the impacts of the COVID-19 pandemic on personal and environmental levels. Regarding the personal level, all forms of in-person sexual contact carry the risk of viral transmission [12]. People may abstain from sexual intercourse because they fear contracting COVID-19. The COVID-19 pandemic may adversely influence individuals’ mental and physical health [2,7,8] and further compromise sex life. Regarding the environmental level, the pandemic prevention measures of social distancing may jeopardize an individual’s sex life. Governments of many countries ordered to close down gathering spaces and promote social distancing to prevent the spread of COVID-19. Not being able to physically meet with others may change people’s sexual habits [14]. A social atmosphere of collective anxiety about the COVID-19 pandemic may also result in detriment of the mood to enjoy sex. Conversely, Yuksel and Ozgor proposed people may spend more time at home during the COVID-19 pandemic and therefore increase the frequency of sexual behaviors with their partners [15].

A few studies have examined the effect of the COVID-19 pandemic on sexual behaviors. A study comparing sexual behaviors among Turkish women during the pandemic with 6–12 months prior to the pandemic found that sexual desire and frequency of intercourse significantly increased during the COVID-19 pandemic, whereas quality of sexual life significantly decreased; the pandemic is associated with decreased desire for pregnancy, decreased female contraception, and increased menstrual disorders [15]. An online study in three south-east Asian countries (Bangladesh, India, and Nepal) found that although 45% of the participants reported that the lockdown affected their sexual life, there was no substantial difference in sexual activity between before and during the lockdown period of the COVID-19 pandemic [16]. However, the study of Yuksel and Ozgor focused on women and did not examine sexual behaviors in men [15]. The sample size in the study of Arafat et al. was small (120 respondents) [16]. Given the importance of sexuality to health, further study is needed to examine how the COVID-19 pandemic has changed peoples’ sex life and what factors are associated with such change [12].

### 1.3. Associations of Risk Perception and General Anxiety with Changes in Sex Life

Risk perception of COVID-19 may influence individuals’ engagement in sexual activity during the pandemic. Risk perception of COVID-19 refers to people’s intuitive evaluations of how likely they are to contract COVID-19 [17]. According to stage theory, risk perception acts as a trigger for precautionary action [18]. Leung et al. reported that people with higher risk perceptions of severe acute respiratory syndrome (SARS) were more likely to take comprehensive precautionary measures against infection [19]. The fear of contagion itself may reduce physical contact within couples, from simple kissing to full sexual intercourse [14]. If the risk perception is excessive, particularly when a person is overwhelmed with negative SARS-related information, it will lead to irrational fear [20].

Psychological disturbances, such as anxiety and depression, may also reduce satisfaction with an individual’s sex life and increase sexual dysfunction [21,22]. The prevalence of anxiety symptoms was high among the public during the SARS [23] and COVID-19 outbreaks [24]. Both risk perception and anxiety can result in psychological stress. Stressful lifestyle is a factor that may negatively impact people’s sexual desire, though the results of previous studies were mixed [25,26,27]. Given that the COVID-19 is a novel respiratory infectious disease resulting in global impacts on human lives worldwide, additional studies are required on whether risk perception of COVID-19 and general anxiety are significantly associated with changes in sex life during the COVID-19 pandemic.

### 1.4. Demographic Factors of Changes in Sex Life

Gender and age have been found to influence the degree of sex-related quality of life. Forbes et al. reported that unlike quality of life in other life domains [28], sex-related quality of life tends to decline with age. Pinxten and Lievens also noted a considerable gender gap in the propensity to engage in sexual activity [29], with this gender gap throughout the life course being smaller in more gender-equal societies. The roles of gender and age in changes in sex life during epidemics of novel infectious diseases, such as SARS, Ebola, or H1N1 influenza, have yet to be examined.

In addition to gender and age, sexual orientation should also be analyzed as an influence on changes in sex life during epidemics. A study hypothesized that in times of a pandemic when people are particularly susceptible to mental health difficulties, messaging that frames sex as being dangerous has adverse psychological effects [12]. There is no evidence that the COVID-19 can be transmitted via either vaginal or anal intercourse; however, there is evidence of oral–fecal transmission of the COVID-19 and that implies that anilingus may represent a risk for infection [30]. In this case, sexual stigma may have a negative role for sex life of sexual minority individuals [12]. From a historical perspective, human immunodeficiency virus and acquired immunodeficiency syndrome (AIDS) was conceived of as a “gay plague,” by analogy with the sudden, devastating epidemics of the past [31]. Considering the historical trauma of the AIDS epidemic, sexual minority individuals may voluntarily or involuntarily restrict their sex life. In addition to sexual stigma, sexual minority individuals may experience more severe impacts of the COVID-19 pandemic on their daily lives. A study on 1051 men who have sex with men (MSM) in the United States found that most of the participants had no change in condom access or use, many participants had adverse impacts of the COVID-19 pandemic to general wellbeing, social interactions, money, food, drug use, and alcohol consumption; especially many reported fewer sex partners and COVID-19-related barriers to HIV and sexually transmitted diseases testing, pre-exposure prophylaxis access, and HIV treatment [32]. The authors also suggested that additional studies of COVID-19 epidemiology among sexual minority populations are needed [32]. Whether the COVID-19 pandemic has affected the sex lives of sexual minority individuals differently than that of their heterosexual counterparts warrants further study.

### 1.5. The Aims of This Study

The online survey study on COVID-19 in Taiwan was conducted to assess the life experiences of people in Taiwan during the COVID-19 outbreak. The survey examined changes in sex life, including satisfaction with the individual’s sex life, frequency of sexual activity, frequency of sex-seeking activity, and frequency of using protection for sex. The present study used data from this survey to examine three issues. First, we compared changes in sex life prior to and during the COVID-19 pandemic. Second, we examined the associations of gender, age, sexual orientation, risk perception of COVID-19, and general anxiety with changes in sex life. Third, we determined what the moderators are in the associations between the aforementioned factors and changes in sex life.

## 2. Methods

### 2.1. Participants

Participants were recruited through a Facebook advertisement from April 10 to 20, 2020. During the study period, the total number of confirmed cases contracting COVID-19 in Taiwan increased from 382 to 422 without no new deaths. Of the 40 newly confirmed cases, only 1 was domestic. It indicated that this study was conducted during the period of COVID-19 mitigation in Taiwan. However, a collective COVID-19 infection among the members of the Navy serving on a three-ship fleet broke out during the period. Meanwhile, the total number of infected cases all over the world massively increased to nearly 800,000. The domestic and foreign situations intensified worry about COVID-19 among people in Taiwan.

Facebook users were eligible for this study if they were at least 20 years of age and living in Taiwan. The Facebook advertisement included a headline, main text, pop-up banner, and link to the research questionnaire website. We designed the advertisement to appear in the Facebook users’ “news feeds,” which is a continually updated list of updates from advertisers and the user’s connections (such as friends and the Facebook groups that they have joined). Our advertisement was placed only in the targeted users’ news feed, rather than other advertising locations (e.g., right-hand column), on Facebook because news feed advertisements are most effective in recruiting research respondents [33]. We targeted the advertisement to Facebook users by location (Taiwan) and language (Chinese), where Facebook’s advertising algorithm determined which users to show our advertisement to. To ensure that sexual minority individuals were recruited, we also posted the link of the Facebook advertisement to the Facebook pages of three Taiwanese health promotion and counseling centers for lesbian, gay, and bisexual individuals.

This study was approved by the Institutional Review Board (IRB) of Kaohsiung Medical University Hospital (KMUHIRB-EXEMPT(I) 20200011). Because participation was voluntary and survey responses were anonymous, the IRB ruled that this study did not require informed consent. Our study respondents were given no incentive for participation. We provided links to COVID-19 information from the Taiwan Centers for Disease Control (Taiwan CDC), Kaohsiung Medical University Hospital, and Medical College of National Cheng Kung University for respondents to learn more about COVID-19.

### 2.2. Measures

The survey comprised the following sections.

#### 2.2.1. Changes in Sex Life during the COVID-19 Pandemic

We measured four aspects of self-reported changes in sex life using four questions, which were adopted from a previous study’s questionnaire on an individual’s sex life [34]. First, change in satisfaction with sex life was asked about in “Compared with that before the COVID-19 outbreak, how has your satisfaction with sex life changed in the past 1 month?” Second, change in sexual activity was asked about in “Compared with that before the COVID-19 outbreak, how has your sexual activity changed in the past 1 month?” Third, change in sex-seeking activity was asked about in “Compared with that before the COVID-19 outbreak, how has your sex-seeking activity, such as using dating apps or visiting a sex worker, changed in the past 1 month?” Fourth, change in using protection for sex was asked about in “Compared with that before the COVID-19 outbreak, how has your use of protection during sex, such as wearing a condom or taking pre-exposure prophylaxis, changed in the past 1 month?” Each question was rated as 0 (obviously decreased), 1 (slightly decreased), 2 (no change), 3 (slightly increased), and 4 (obviously increased).

#### 2.2.2. Risk Perception of COVID-19

To measure risk perception of COVID-19, we used the 5-item questionnaire that was developed by Liao and colleagues to measure worry toward H1N1 influenza [35]. The first question was “If you were to develop flu-like symptoms tomorrow, you would be ___,” where respondents filled in the blank with a number from 1 (not at all worried) to 5 (extremely worried). The second question was “In the past 1 week, have you ever worried about getting COVID-19?” This question was scored from 1 (no, I have never thought about it) to 5 (I worry about it all the time). The third question was “Please rate the current level of your worry toward COVID-19.” This question was rated from 1 (very mild) to 10 (very severe). The fourth question was “How likely do you think it is that you will contract COVID-19?” This question was scored from 1 (never) to 7 (certain). The fifth question was “What do you think are your chances of getting COVID-19 over the next 1 month compared with others outside your family?” This question was rated from 1 (not at all) to 7 (certain). The scores for the first two, the third, and the last two questions were divided by 5, 10, and 7, respectively. These five quotients were then summed to obtain a score representing the level of risk perception of COVID-19, with higher scores indicating greater risk perception. Cronbach’s α was 0.759 for this measure.

#### 2.2.3. General Anxiety

General anxiety in the past 1 week was measured using a previously validated state anxiety scale from the State-Trait Anxiety Inventory wherein respondents rate their feelings in response to 10 general statements; these statements inquire into feelings of being rested, contented, comfortable, relaxed, pleasant, anxious, nervous, jittery, highly strung, and over-excited and “rattled” [35,36,37]. Each question was rated as 1 (not at all), 2 (sometimes), 3 (moderately so), and 4 (very much so). Statements reflecting positive feelings were reversely coded. A higher total score of the 10 items represented greater general anxiety. The Cronbach’s α was 0.921 for this measure.

#### 2.2.4. Demographic Variables

Data on sexual orientation (heterosexual, homosexual, bisexual, pansexual, asexual, or unsure), gender (female and male), and age were collected. Respondents were categorized into heterosexual and sexual minority individuals.

### 2.3. Statistical Analysis

Data analysis was performed using the statistical software SPSS (version 22.0; SPSS Inc., Chicago, IL, USA). We noted the percentages of participants reporting no change, an increase, and a decrease for changes in each aspect of their sex life. The associations of changes in each aspect of sex life with sexual orientation, gender, age, risk perception of COVID-19, and general anxiety were examined using multivariate logistic regression. The *p* value, odds ratio (OR), and 95% confidence interval (CI) were used to indicate significance. A two-tailed *p* value of <0.05 indicated statistical significance. Moreover, the moderators of the associations between sexual orientation and changes in sex life were examined based on the criteria proposed by Baron and Kenny [38].

## 3. Results

### 3.1. Changes in Sex Life

The data of 533 sexual minority and 1421 heterosexual individuals were analyzed, with 77 of the initial 2031 respondents excluded due to missing data. Among sexual minority respondents, 320 identified as homosexual, 164 identified as bisexual, and 49 identified as pansexual, asexual, or unsure. Table 1 presents descriptive statistics for demographic characteristics, risk perception of COVID-19, general anxiety, and changes in sex life among respondents. The mean age was 37.9 years (standard deviation [SD] = 10.8 years); and 66.8% were female. The mean scores for risk perception of COVID-19 and general anxiety were 2.9 (SD = 0.7) and 23.1 (SD = 6.6), respectively.

As for changes in sex life, respondents reported decreases in their satisfaction with their sex life (13.4%), their sexual activity (13.5%), their sex-seeking activity (6.7%), and their use of protection for sex (1.4%) during the COVID-19 pandemic. By contrast, respondents reported increases in their satisfaction with their sex life (1.9%), their sexual activity (2.9%), their sex-seeking activity (1.9%), and their use of protection for sex (1.6%) during the COVID-19 pandemic.

### 3.2. Factors Relating to Changes in Sex Life

Table 2 presents the multivariate logistic regression results for the associations of changes in each aspect of sex life (dependent variables) with sexual orientation, gender, age, risk perception of COVID-19, and general anxiety (independent variables). The results indicated that decreased satisfaction with sex life was significantly associated with being male, being a sexual minority, and having a higher level of general anxiety; no factor was significantly associated with increased satisfaction with sex life. Regarding frequency of sexual activity, decreased frequency was significantly associated with being male, being a sexual minority, having a higher risk perception of COVID-19, and having greater general anxiety; increased frequency was significantly associated with being male, being younger, and having a lower risk perception of COVID-19.

Regarding frequency of sex-seeking activity, decreased frequency was significantly associated with being male, being a sexual minority, having a higher risk perception of COVID-19, and having greater general anxiety; increased frequency was significantly associated with being male and being a sexual minority. Finally, being male was significantly associated with both decreased and increased frequency of the use of protection for sex.

### 3.3. Moderators of the Associations Between Related Factors and Changes in Sex Life

The product interaction terms between gender, age, sexual orientation, risk perception, and general anxiety that were significantly associated with each aspect of change in sex life were selected into multiple logistic regression models to examine the moderators (Table 3). The results indicated that the interaction between gender and sexual orientation was significantly associated with decreased frequencies of sexual activity and sex-seeking activity. Further analysis revealed that the significant association between being sexual minority and decreased frequency of sexual activity was only true for men (OR = 2.466, 95% CI: 1.573–3.868, *p* < 0.001) but not for women (OR = 1.098, 95% CI: 0.690–1.748, *p* = 0.692). Moreover, the significant association between being a sexual minority and decreased frequency of sex-seeking activity was only true for men (OR = 7.281, 95% CI: 3.975–13.339, *p* < 0.001) but not for women (OR = 1.983, 95% CI: 0.988–3.978, *p* = 0.054).

## 4. Discussion

### 4.1. Changes in Sex Life

Most respondents reported no change in their sex life during the COVID-19 pandemic, whereas 13.4% reported the decrease satisfaction with their sex life, 13.5% decreased sexual activity, 6.7% decreased sex-seeking activity, and 1.6% reported the increased use of protection for sex. By contrast, 1.9% reported the increased satisfaction with their sex life, 2.9% increased sexual activity, 1.9% increased sex-seeking activity, and 1.4% reported the decreased use of protection for sex. Sexuality is a central aspect of human health [12]. Considering the ineffectiveness of recommendations of long-term sexual abstinence [39], health care providers should consider counseling people on their sexual health whenever possible to help them maintain or even improve their sexual wellness amid the COVID-19 pandemic; in general, the human need for intimacy should be balanced with personal safety and pandemic control [12]. Further study is also warranted to understand the psychological mechanisms underlying sexuality during a pandemic. Although only 1.4% of respondents reported the decreased use of protection for sex in the present study, the increased risk of contracting COVID-19 and sexually transmitted diseases should also be investigated in people who engage in sexual activity more but use protection less during the pandemic.

### 4.2. Factors Relating to Changes in Sex Life

The present study found that risk perception, general anxiety, gender, age, and sexual orientation related to various aspects of changes in sex life during the COVID-19 pandemic. Firstly, a higher risk perception of COVID-19 was significantly associated with decreased frequencies of sexual activity and sex-seeking activities. Because in-person sexual contact carries the risk of COVID-19 transmission [12], we can hypothesize that people with a high risk perception of COVID-19 reduce their sexual activity, whether casual or with long-term partner, as a means to protect themselves against getting COVID-19. However, Leung et al. noted that the self-perceived likelihood of contracting or surviving SARS did not predict personal protective behavior [36]. In addition to sexual inactivity as a personal protective behavior, other reasons may also account for the association between risk perception of COVID-19 and the decreased frequencies of sexual activity and sex-seeking activities. People who perceive a high risk of COVID-19 tend to invest greater time and effort to learning more about pandemic prevention and using facemasks and disinfectant alcohol. Moreover, as with governments in many other countries, the Taiwanese government has suspended the sex industry to prevent the spread of COVID-19. This policy may have led people with a high risk perception of COVID-19 to perceive having sex in general as being unsafe during the pandemic. These personal and environmental factors may have contributed to the decreased frequencies of sexual activity and sex-seeking activities during the pandemic.

The present study found that higher general anxiety was significantly associated with decreased satisfaction of sex life and frequencies of sexual activity and sex-seeking activities during the COVID-19 pandemic. Greater general anxiety may result in less pleasurable sex or make sex difficult, which depresses a person’s interest in sex; sexual dysfunction may further exacerbate anxiety [40]. However, anxiety is one of most common affective responses to the respiratory infectious disease epidemics or pandemics [41]. During the respiratory infectious disease outbreaks, social distancing and quarantine are inevitable methods to prevent spreading of the illness; however, both may precipitate anxiety [42]. Thus, health care providers should help people manage their anxiety in particular and mental health in general.

A study on Turkish women found that despite the increased frequency of sexual intercourse during the COVID-19 pandemic, quality of sexual life decreased during the pandemic [15]. The present study extends the scope of genders in participants and found that men were more likely to report a change, whether an increase or decrease, in their frequencies of sexual activity, sex-seeking activities, and use of protection for sex. A UK study also noted that men in the general population were more likely to engage in sexual activity during COVID-19 self-isolation [27]. Both biological and socioenvironmental factors may influence sexual behaviors across genders. Research on 3- to 6-year-old children found significant gender differences in sexual behavior [43]. Gender stereotypes in societies may also affect the way people behave in sexual activities [44,45]. There might be multiple etiologies accounting for gender difference in changes of sexual activities during the COVID-19 pandemic that warrant further study. Moreover, research in China has demonstrated that women tend to be more psychologically affected by the COVID-19 outbreak with respect to stress, anxiety, depression, and posttraumatic stress symptoms [46,47]. The results of previous and present study indicated that the COVID-19 pandemic might have various impacts on sexual activities differently compared with and psychological wellbeing, as well as that changes in sex life might not be influenced by psychological status only.

Younger people were noted to be at greater risk of mental illnesses, such as general anxiety disorder, during the COVID-19 pandemic [24]. Consistent with the aforementioned study, the present study found that older respondents were less likely to have increased sexual activity during COVID-19 pandemic [48]. Gender and age differences in sexual activity have already been noted by a Taiwanese study before the pandemic; these differences include men being more likely to have multiple sexual partners and older people being more likely to have a lower frequency of sexual activity [49]. The results of the present study demonstrate that the COVID-19 pandemic has had differential effects on sex life across gender and age. The factors that explain the reason that men are more likely to have a pandemic-induced change in their sex life—whether an increase or decrease in the aspects analyzed in this study—should be investigated further.

The present study found that sexual minority respondents were more likely to report decreased satisfaction with their sex life and decreased frequencies of sexual activity and sex-seeking activities during the pandemic; a gender difference with respect to the aforementioned frequencies was also noted. Sexual minority individuals have faced the threat of HIV infection since the 1980s [50], and health information pertaining to HIV prevention and treatment strategies is transferable to knowledge on COVID-19 [51]. Thus, being more knowledgeable about pandemic prevention, sexual minority individuals may reduce their sexual activity, albeit at the expense of reducing their satisfaction with their sex life. However, the decreases in sexual activity could also be explained by sexual stigma, which sexual minority communities have experienced since the AIDS pandemic [52].

Recommendations for sexual abstinence during the COVID-19 pandemic may elicit memories of the widespread stigmatization of sexual minorities during the AIDS crisis [12]. Such sexual stigma contributes to a hostile social environment against sexual minority individuals and makes mental health problems more likely [53]. Sexual minority individuals may reduce their sexual activity under the interpersonal strain caused by the COVID-19 pandemic. However, a group of sexual minority individuals have increased their sex-seeking activity during the COVID-19 pandemic. Whether this is a response to the unhappy atmosphere during the pandemic warrants further study.

### 4.3. Limitations

The present study has some limitations. First, the present study did not ask participants about their current sexual relationship status or relationship quality, which are major factors affecting people’s sexual satisfaction during a pandemic like Covid-19 where people’s social lives may have been curtailed. Sex between intimate couples can be an activity to support psychologically fragile people living in restricted areas for longer quarantine periods [54]. Second, although recruiting respondents through Facebook is a promising research method for targeting the general public during fast-moving infectious disease outbreaks [55], Facebook users may not be representative of the population. A review of a study that recruited respondents through Facebook reported a bias in favor of women, young adults, and people with higher education and incomes [56]. Third, the cross-sectional design of this study limited causal inference between changes in sex life and general anxiety. Fourth, this study did not survey the various aspects of respondents’ sex life before the pandemic. This study also did not follow-up on the changes in respondents’ sex life during the mitigating period of the COVID-19 pandemic. Last, some factors such as physical health and self-confidence that might influence sex life in the COVID-19 pandemic were not examined in the present study.

## 5. Conclusions

Although most respondents reported no change in their sex life during the COVID-19 pandemic, 1.4%–13.5% of respondents reported a decrease and 1.6%–2.9% reported an increase in various aspects of their sex life. Risk perception of COVID-19, general anxiety, gender, age, and sexual orientation were associated with greater change in various aspects of an individual’s sex life. Considering that sexuality is a central aspect of human health, human needs for intimacy should be balanced with personal safety and pandemic control [12]. Health care providers should consider the factors related to changes in sex life reported in this study when formulating strategies for maintaining sexual wellness amid respiratory infection epidemics.

## Figures and Tables

**Table 1 ijerph-17-05822-t001:** Comparisons of Demographic Characteristics, Risk Perception of COVID-19, General Anxiety, and Changes in Sex Life Between Heterosexual and Sexual Minority Respondents (N = 1954).

Variables	*n* (%)	Mean (SD)	Range
Gender			
Female	1305 (66.8)		
Male	649 (33.2)		
Age (years)		37.9 (10.8)	20–74
Sexual orientation			
Heterosexual	1421 (72.7)		
Sexual minority	533 (28.3)		
Risk perception of COVID-19		2.9 (0.7)	0.8–5
General anxiety		23.1 (6.6)	10–40
Changes in sexual life			
Satisfaction with sexual life			
Unchanged	1655 (84.7)		
Decreased	261 (13.4)		
Increased	38 (1.9)		
Frequency of sexual activity			
Unchanged	1633 (83.6)		
Decreased	264 (13.5)		
Increased	57 (2.9)		
Frequency of sex-seeking activity			
Unchanged	1786 (91.4)		
Decreased	131 (6.7)		
Increased	37 (1.9)		
Frequency of protection for sex			
Unchanged	1894 (96.9)		
Decreased	28 (1.4)		
Increased	32 (1.6)		

COVID-19: Coronavirus Disease 2019; SD: standard deviation.

**Table 2 ijerph-17-05822-t002:** Associations of Gender, Age, Sexual Orientation, Risk Perception of COVID-19, and General Anxiety with Changes in Sex Life.

	Satisfaction with Sexual Life				Frequency of Sexual Activity				Frequency of Sex-seeking Activity				Frequency of Protection for Sex			
	Decreased ^a^		Increased ^a^		Decreased ^a^		Increased ^a^		Decreased ^a^		Increased ^a^		Decreased ^a^		Increased ^a^	
	aOR (95% CI)	*p*	aOR (95% CI)	*p*	aOR (95% CI)	*p*	aOR (95% CI)	*p*	aOR (95% CI)	*p*	aOR (95% CI)	*p*	aOR (95% CI)	*p*	aOR (95% CI)	*p*
Males ^c^	1.860 (1.392–2.484)	<0.001	0.971 (0.479–1.969)	0.935	2.463 (1.857–3.267)	<0.001	1.955 (1.118–3.418)	0.019	4.191 (2.794–6.285)	<0.001	2.133 (1.079–4.216)	0.029	4.091 (1.841–9.090)	0.001	4.510 (1.949–10.434)	<0.001
Age	0.993 (0.980–1.007)	0.339	0.995 (0.963–1.027)	0.743	1.003 (0.989–1.016)	0.691	0.966 (0.938-0.993)	0.016	1.007 (0.988–1.027)	0.453	0.994 (0.960–1.029)	0.727	1.033 (0.999–1.068)	0.056	0.997 (0.960–1.035)	0.869
Sexual minority ^b^	1.837 (1.349–2.503)	<0.001	1.461 (0.699–3.056)	0.314	1.666 (1.224–2.267)	0.001	1.343 (0.745–2.422)	0.327	4.245 (2.770–6.505)	<0.001	2.718 (1.314–5.625)	0.007	1.983 (0.860–4.573)	0.108	1.264 (0.531–3.006)	0.597
Risk perception of COVID-19	1.225 (0.981–1.530)	0.073	0.760 (0.453–1.274)	0.298	1.275 (1.024–1.589)	0.030	0.607 (0.396-0.929)	0.022	1.428 (1.042–1.956)	0.026	0.785 (0.464–1.32)	0.367	1.745 (0.966–3.149)	0.065	1.084 (0.585–2.008)	0.799
General anxiety	1.078 (1.054–1.103)	<0.001	0.957 (0.903–1.014)	0.140	1.043 (1.020–1.066)	<0.001	1.014 (0.970–1.060)	0.531	1.047 (1.015–1.080)	0.004	1.022 (0.968–1.079)	0.433	1.046 (0.986–1.110)	0.136	1.023 (0.961–1.089)	0.477

^a^ Unchanged as reference; ^b^ Heterosexual as reference; ^c^ Female as reference. COVID-19: Coronavirus Disease 2019. aOR: adjusted odds ratio; CI: confidence interval.

**Table 3 ijerph-17-05822-t003:** Moderators of Associations Between Gender, Age, Sexual Orientation, Risk Perception, and General Anxiety with Changes in Sex Life ^a.^

	Satisfaction with Sexual Life		Frequency of Sexual Activity				Frequency of Sex-seeking Activity			
	Decreased ^b^		Decreased ^b^		Increased ^b^		Decreased ^b^		Increased ^b^	
	aOR (95% CI)	*p*	aOR (95% CI)	*p*	aOR (95% CI)	*p*	aOR (95% CI)	*p*	aOR (95% CI)	*p*
Gender X Age					0.980 (0.928–1.035)	0.469				
Gender X Sexual orientation	1.335 (0.724–2.463)	0.354	2.238 (1.214–4.125)	0.010			3.102 (1.319–7.292)	0.009	2.346 (0.576–9.545)	0.234
Gender X Risk perception of COVID-19			0.723 (0.455–1.147)	0.168	1.586 (0.725–3.469)	0.248	.979 (0.486–1.970)	0.952		
Gender X General anxiety	1.006 (0.964–1.050)	0.781	1.022 (0.974–1.072)	0.372			1.037 (0.967–1.112)	0.312		
Age X Risk perception of COVID-19					0.994 (0.957–1.033)	0.772				
Sexual orientation X Risk perception of COVID-19			1.027 (0.625–1.686)	0.917			0.898 (0.454–1.774)	0.756		
Sexual orientation X General anxiety	0.993 (0.980–1.006)	0.273	1.034 (0.984–1.087)	0.180			1.036 (0.967–1.110)	0.318		

^a^ Controlling for gender, age, sexual orientation, risk perception, and general anxiety. ^b^ Unchanged as reference. COVID-19: Coronavirus Disease 2019. aOR: adjusted odds ratio; CI: confidence interval.

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
