# Peer review of "Changes in Sex Life among People in Taiwan during the COVID-19 Pandemic: The Roles of Risk Perception, General Anxiety, and Demographic Characteristics"

_ijerph, 2020, doi:10.3390/ijerph17165822_

Round 1
Reviewer 1 Report
This paper reports on a timely, interesting and well-designed study. The data collection and analysis methods are sound and appropriate to the study. However, the paper needs some revisions to bring it to publishable standard.
The introduction needs substantial revisions. It is difficult to follow the rationale for the study. The first paragraph needs to more clearly explain why we might expect changes in people’s sexual lives due to Covid-19 and put a timeline to this. Eg. Was there a period of social lockdown in Taiwan that would have affected social lives? What were case numbers of Covid-19 in Taiwan relative to other regions and countries? The introduction seems to imply that we might expect a reduction in sexual activity/sexual connection due to Covid-19 but this is not explicitly stated. Also, the introduction states that “sexual health in the context of a pandemic warrants investigation”, but what aspects of sexual health are under investigation in this study and why? Why might be expect changes in sexual health behaviour or STI screening during a pandemic?
The subsection titled “Factors related to the Change in Sex Life” also needs revision to enhance clarity. I would suggest creating two subsections: the first specifically related to Covid-19 and why we might expect Covid-19 to effect sex lives. This section may refer to factors such as enhanced anxiety or fear of contagion. The second subsection should look at other factors known to effect sexual satisfaction. This needs to be more comprehensively research and referenced than it is at present. At present, there are only two references and a major focus on age and life-course, which makes no sense to the study. One broad statement about age and sexual satisfaction (which doesn’t actually accord with a lot of the literature) ignores the bulk of research that shows factors affecting sexual quality of life tend to be relationship quality, physical health, anxiety and self-confidence.
The rationale for an extra focus on sexual minorities due to a likelihood of heightened vulnerability to sexual stigma needs more explaining. I agree with the authors that sexual minorities (and gay and bisexual men in particular) face extra scrutiny of their sexual practices – often based on the negative stereotypes about gay and bisexual men that mark HIV-related stigma. However, to make this argument work, the authors need to explain a bit more about why we might expect to see greater scrutiny or surveillance over people’s lives – and sex lives in particular – due to Covid-19.
The study methods are well explained and clear. However, there does need to be more information about the timing of Covid-19 relative to recruitment. Between 10 and 20 April, what was the status of Covid-19 in Taiwan?
The results are difficult to follow in parts. I would suggest the following changes:
- Begin with a description of the key characteristics of the sample.
- Listing percentages makes the wording in the first paragraph makes it very difficult to follow. It would be more insightful and useful for the reader to state the most significant findings in each section (ie. that the majority reported no changed in all categories).
The discussion is well written and draws together key points from the findings in a useful way. It does seem to be an important finding that anxiety was related to a reduced level of sex and sexual satisfaction. This accords with previous studies on sexual satisfaction but also likely points to the impact of Covid-19 on people’s lives. It is likely that there was an increase in general anxiety due to Covid-19 and this had an impact on people’s everyday lives, including their sexual lives. The authors could highlight this findings more.
I think a major limitation of this study was that it did not (as far as I can tell) ask people about their current sexual relationship status or relationship quality. These are major factors affecting people’s sexual satisfaction and presumably more so during a pandemic like Covid-19 where people's social lives may have been curtailed. This is a large gap in a study that seeks to understand people’s sex lives and should be noted.
Author Response
Comment 1-1
The introduction needs substantial revisions. It is difficult to follow the rationale for the study. The first paragraph needs to more clearly explain why we might expect changes in people’s sexual lives due to Covid-19 and put a timeline to this. Eg. Was there a period of social lockdown in Taiwan that would have affected social lives? What were case numbers of Covid-19 in Taiwan relative to other regions and countries? The introduction seems to imply that we might expect a reduction in sexual activity/sexual connection due to Covid-19 but this is not explicitly stated.
Response
Thank you for your valuable comments. We added a new paragraph introducing the situation of COVID-19 pandemic in Taiwan including the number of confirmed cases and policies of control COVID-19 infection spreading into Introduction section as below. Please refer line 40-61.
“1.1. The Coronavirus Disease 2019 (COVID-19) Pandemic in Taiwan
The COVID-19 emerged in Wuhan, China, at the end of 2019 has been spread to over 200 countries and territories around the world, with the total number of infected cases have risen to over 14,000,000 and more than 590,000 deaths, as of 17 July 2020 [1]. Given geographical proximity and busy transportation between Taiwan and China, Taiwan was predicted to have the second highest number of COVID-19 cases early in the outbreak [2]. After all, Taiwan experienced the severe outbreak of 2002–2003 Severe Acute Respiratory Syndrome (SARS), which also originated from China; globally, Taiwan had the third highest number of SARS cases, after China and Hong Kong [3]. The experience with SARS made Taiwanese government and people vigilant against COVID-19. Early since December 31, 2019, Taiwanese officials began to assess passengers on direct flights from Wuhan for fever and pneumonia symptoms before passengers could deplane.
The first COVID-19 case In Taiwan was confirmed on January 21, 2020 [4]. During the period from January 20 to February 24, the Taiwan Centers for Disease Control has rapidly produced and implemented a list of at least 124 action items including border control, case identification, quarantine of suspicious cases, proactive case finding, resource allocation, reassurance and education of the public while fighting misinformation, negotiation with other countries and regions, formulation of policies toward schools and childcare, and relief to businesses [5]. With proactive containment efforts and comprehensive contact tracing, the number of COVID-19 cases in Taiwan remained low, as compared with other countries that had widespread outbreaks [6]. Therefore, there was no social lockdown in Taiwan. As of 17 July 2020, Taiwan had tested a total of 79,395 persons showing 454 confirmed cases, of which only 55 were domestic. Six patients had died, and 440 people had been released from hospital after testing negative three times subsequently [1].”
Comment 1-2
Also, the introduction states that “sexual health in the context of a pandemic warrants investigation”, but what aspects of sexual health are under investigation in this study and why? Why might be expect changes in sexual health behaviour or STI screening during a pandemic?
Response
We revised the contents of Introduction section based on the suggestion of the World Health Organization for sexual health and sexuality to illustrate our aspect of study on change of sex life during the COVID-19 pandemic as below. We also summarized the results of two studies on sexual behaviors during the COVID-19 pandemic to support the necessity of further study on sex life during the COVID-19 pandemic as below.
“According to the World Health Organization, sexual health is a state of physical, mental, and social well-being in relation to sexuality [13]; sexuality is influenced by the interaction of biological, psychological, social, economic, political, cultural, legal, historical, religious and spiritual factors [13]. Accordingly, it is reasonable to hypothesize that sex life would be deeply influenced by the impacts of the COVID-19 pandemic on personal and environmental levels. Regarding to the personal level, all forms of in-person sexual contact carry the risk of viral transmission [12]. People may abstain from sexual intercourse because they fear contracting COVID-19. The COVID-19 pandemic may adversely influence the individuals’ mental and physical health [2,7,8] and further compromise sex life. Regarding to the environmental level, the pandemic prevention measures of social distancing may jeopardize an individual’s sex life. Governments of many countries ordered to close down gathering spaces and promote social distancing to prevent the spread of COVID-19. Not being able to physically meet with others may change people’s sexual habits [14]. Social atmosphere of collective anxiety about the COVID-19 pandemic may also result in detriment of mood to enjoy sex. Conversely, Yuksel and Ozgor proposed people may spend more time at home during the COVID-19 pandemic and therefore increase the frequency of sexual behaviors with their partners [15].” Please refer line 66-80.
“A few studies have examined the effect of the COVID-19 pandemic on sexual behaviors. A study comparing sexual behaviors among Turkish women during the pandemic with 6-12 months prior to the pandemic found that sexual desire and frequency of intercourse significantly increased during the COVID-19 pandemic, whereas quality of sexual life significantly decreased; the pandemic is associated with decreased desire for pregnancy, decreased female contraception, and increased menstrual disorders [15]. An online study in three south-east Asian countries (Bangladesh, India & Nepal) found that although 45% of the participants reported that the lockdown affected their sexual life, there was no substantial difference in sexual activity between before and during the lockdown period of the COVID-19 pandemic [16]. However, the study of Yuksel and Ozgor focused on women and did not examine sexual behaviors in men [15]. The sample size in the study of Arafat et al. was small (120 respondents) [16]. Given the importance of sexuality to health, further study is needed to examine how the COVID-19 pandemic has changed peoples’ sex life and what factors are associated with such change [12].” Please refer line 81-92.
Comment 2
The subsection titled “Factors related to the Change in Sex Life” also needs revision to enhance clarity. I would suggest creating two subsections: the first specifically related to Covid-19 and why we might expect Covid-19 to effect sex lives. This section may refer to factors such as enhanced anxiety or fear of contagion. The second subsection should look at other factors known to effect sexual satisfaction. This needs to be more comprehensively research and referenced than it is at present. At present, there are only two references and a major focus on age and life-course, which makes no sense to the study. One broad statement about age and sexual satisfaction (which doesn’t actually accord with a lot of the literature) ignores the bulk of research that shows factors affecting sexual quality of life tend to be relationship quality, physical health, anxiety and self-confidence.
Response
Thank you for your comment. In the revised manuscript we divided the paragraph “Factors related to the Change in Sex Life” into two paragraphs titled ““1.3. Associations of Risk Perception and General Anxiety with Changes in Sex Life” and “1.4. Demographic Factors of Changes in Sex Life” in Introduction section (line 93 and 110). We also added new contents based on the results of previous studies on the associations of risk perception and anxiety with changes in sex life in sections 1.3. and 1.4. as below. We agreed that there might be factors other than risk perception, general anxiety and demographic characteristics related to changes in sex life in the COVID-19 pandemic but not examined in the present study. Therefore, we revised our topic of study and focused on “The Roles of Risk Perception, General Anxiety, and Demographic Characteristics” (please refer to line 3-4). We also listed it as one of limitations of this study as below.
“The fear of contagion itself may reduce physical contact within couples, from simple kissing to full sexual intercourse [14].” Please refer line 99-100.
“Both risk perception and anxiety can result in psychological stress. Stressful lifestyle is a factor that may negatively impact people’s sexual desire, though the results of previous studies were in mixed [25-27]. Given that the COVID-19 is a novel respiratory infectious disease resulting in global impacts on human lives worldwide, additional studies are required on whether risk perception of COVID-19 and general anxiety are significantly associated with changes in sex life during the COVID-19 pandemic.” Please refer line 104-109.
“A study on 1051 men who have sex with men (MSM) in the United States found that most of the participants had no change in condom access or use, many participants had adverse impacts of the COVID-19 pandemic to general wellbeing, social interactions, money, food, drug use and alcohol consumption; especially many reported fewer sex partners and COVID-19 related barriers to HIV and sexually transmitted diseases testing, pre-exposure prophylaxis access, and HIV treatment [32]. The authors also suggested that additional studies of COVID-19 epidemiology among sexual minority population are needed [32].” Please refer line 128-134.
“Last, some factors such as physical health and self-confidence that might influence sex life in the COVID-19 pandemic were not examined in the present study.” Please refer line 367-369.
Comment 3
The rationale for an extra focus on sexual minorities due to a likelihood of heightened vulnerability to sexual stigma needs more explaining. I agree with the authors that sexual minorities (and gay and bisexual men in particular) face extra scrutiny of their sexual practices – often based on the negative stereotypes about gay and bisexual men that mark HIV-related stigma. However, to make this argument work, the authors need to explain a bit more about why we might expect to see greater scrutiny or surveillance over people’s lives – and sex lives in particular – due to Covid-19.
Response
Thank you for your comment. We revised the paragraph about the possible impacts of the COVID-19 pandemic on sex life of sexual minority individuals as below. Please refer line 118-134.
“A study hypothesized that in times of a pandemic when people are particularly susceptible to mental health difficulties, messaging that frames sex as being dangerous has adverse psychological effects [12]. There is no evidence that the COVID-19 can be transmitted via either vaginal or anal intercourse; however, there is evidence of oral-fecal transmission of the COVID-19 and that implies that anilingus may represent a risk for infection [30]. In this case, sexual stigma may have a negative role for sex life of sexual minority individuals [12]. From a historical perspective, human immunodeficiency virus and acquired immunodeficiency syndrome (AIDS) was conceived of as a "gay plague," by analogy with the sudden, devastating epidemics of the past [31]. Considering the historical trauma of the AIDS epidemic, sexual minority individuals may voluntarily or in voluntarily restrict their sex life. In addition to sexual stigma, sexual minority individuals may experience more severe impacts of the COVID-19 pandemic on their daily lives. A study on 1051 men who have sex with men (MSM) in the United States found that most of the participants had no change in condom access or use, many participants had adverse impacts of the COVID-19 pandemic to general wellbeing, social interactions, money, food, drug use and alcohol consumption; especially many reported fewer sex partners and COVID-19 related barriers to HIV and sexually transmitted diseases testing, pre-exposure prophylaxis access, and HIV treatment [32]. The authors also suggested that additional studies of COVID-19 epidemiology among sexual minority population are needed [32].”
Comment 4
The study methods are well explained and clear. However, there does need to be more information about the timing of Covid-19 relative to recruitment. Between 10 and 20 April, what was the status of Covid-19 in Taiwan?
Response
Thank you for your comment. We added introduction for the status of Covid-19 in Taiwan between 10 and 20 April as below. Please refer line 148-154.
“During the study period, the total number of confirmed cases contracting COVID-19 in Taiwan increased from 382 to 422 without no new deaths. Of the 40 newly confirmed cases, only 1 was domestic. It indicated that this study was conducted during the period of COVID-19 mitigation in Taiwan. However, the collective COVID-19 infection among the members of the Navy serving on a three-ship fleet outbroke during the period. Meanwhile, the total number of infected cases all over the world massively increased for nearly 800,000. The domestic and foreign situations intensified worry about COVID-19 among people in Taiwan.”
Comment 5
The results are difficult to follow in parts. I would suggest the following changes:
- Begin with a description of the key characteristics of the sample.
- Listing percentages makes the wording in the first paragraph makes it very difficult to follow. It would be more insightful and useful for the reader to state the most significant findings in each section (ie. that the majority reported no changed in all categories).
Response
5-1 Thank you for your comment. We revised the manuscript by beginning with a description of the key characteristics of the sample as below. Please refer line 230-232.
“The mean age was 37.9 years (standard deviation [SD] = 10.8 years); and 66.8% were female. The mean scores for risk perception of COVID-19 and general anxiety were 2.9 (SD = 0.7) and 23.1 (SD = 6.6), respectively.”
5-2 We edited the sentences for the rate of changes in sex life in the following way.
“As for changes in sex life, respondents reported decreases in their satisfaction with their sex life (13.4%), their sexual activity (13.5%), their sex-seeking activity (6.7%), and their use of protection for sex (1.4%) during the COVID-19 pandemic. By contrast, respondents reported increases in their satisfaction with their sex life (1.9%), their sexual activity (2.9%), their sex-seeking activity (1.9%), and their use of protection for sex (1.6%) during the COVID-19 pandemic.” Please refer line 233-237.
“13.4% reported the decrease satisfaction with their sex life, 13.5% decreased sexual activity, 6.7% decreased sex-seeking activity, and 1.6% reported the increased use of protection for sex. By contrast, 1.9% reported the increased satisfaction with their sex life, 2.9% increased sexual activity, 1.9% increased sex-seeking activity, and 1.4% reported the decreased use of protection for sex.” Please refer line 277-280.
Comment 6
The discussion is well written and draws together key points from the findings in a useful way. It does seem to be an important finding that anxiety was related to a reduced level of sex and sexual satisfaction. This accords with previous studies on sexual satisfaction but also likely points to the impact of Covid-19 on people’s lives. It is likely that there was an increase in general anxiety due to Covid-19 and this had an impact on people’s everyday lives, including their sexual lives. The authors could highlight the findings more.
Response
Thank you for your comment. We revised this paragraph to highlight the association of general anxiety with changes in sex life. Please refer to line 308-313.
“Greater general anxiety may result in less pleasurable sex or make sex difficult, which depresses a person’s interest in sex; sexual dysfunction may further exacerbate anxiety [40]. However, anxiety is one of most common affective responses to the respiratory infectious disease epidemics or pandemics [41]. During the respiratory infectious disease outbreaks, social distancing and quarantine are inevitable methods to prevent spreading of the illness; however, both may precipitate anxiety [42]. Thus, health care providers should help people manage their anxiety in particular and mental health in general.”
Comment 7
I think a major limitation of this study was that it did not (as far as I can tell) ask people about their current sexual relationship status or relationship quality. These are major factors affecting people’s sexual satisfaction and presumably more so during a pandemic like Covid-19 where people's social lives may have been curtailed. This is a large gap in a study that seeks to understand people’s sex lives and should be noted.
Response
Thank you for your reminding. We listed it as one of the limitations of this study as below. Please refer line 356-360.
“First, the present study did not ask participants about their current sexual relationship status or relationship quality, which are major factors affecting people’s sexual satisfaction during a pandemic like Covid-19 where people's social lives may have been curtailed. Sex between intimate couples can be an activity to support psychologically fragile people living in restricted areas for longer quarantine periods [54].”
Reviewer 2 Report
- “The survey inquired about changes in sex life during the pandemic, including satisfaction with an individual’s sex life, frequency of sexual activity, frequency of sex-seeking activity, and frequency of using protection for sex”
- “ Considering the importance of sexuality to health, sexual counseling and risk reduction are aided by an understanding of how the pandemic has changed peoples’ sex life and what factors are associated with such change” – could be restructured to be clearer.
- “2. Factors Related to the Change in Sex Life” – There are likely multiple changes, so I think it would be better to rephrase the title as “Factors relating [or related] to the changes in sex life”
- I think additional clarity as to why you think a person’s sexual orientation would play a role in changes in sex during COVID-19 pandemic is necessary. You note: “In addition to gender and age, sexual orientation should also be analyzed as an influence on changes in sex life during epidemics. A study hypothesized that in times of a pandemic when people are particularly susceptible to mental health difficulties, messaging that frames sex as being dangerous has adverse psychological effects [7]. In this case, sexual minority individuals may be particularly vulnerable to sexual stigma, considering the historical trauma that sexual minorities have faced in other pandemics, such as that of AIDS [7].” In particular, the last sentence needs to be backed up further.
- “Facebook users were eligible for this study if they were aged ≥20 years and living in Taiwan.” – If they were at least 20 years of age, would fit better in this sentence.
- “Scores of 0 or 1 indicated a decrease, a score of 2 indicated no change, and scores of 3 or 4 indicated an increase.” I believe this sentence is not necessary, as this information is incorporated in the previous sentence.
- “The Cronbach’s α for this section was 0.759” – for this measure, would fit better in this sentence.
- “As for changes in sex life, 13.4%, 13.5%, 6.7%, and 1.4% of respondents reported decreases in their satisfaction with their sex life, their sexual activity, their sex-seeking activity, and their use of protection for sex during the COVID-19 pandemic, respectively” could be improved by editing the sentence in the following way: “Respondents reported decreases in their satisfaction with their sex life (13.4%), their sexual activity (13.5%), their sex-seeking activity (6.7%), and their use of protection for sex (1.4%) during the COVID-19 pandemic” or something along these lines.
- “By contrast, 1.9%, 2.9%, 1.9%, and 1.6% of respondents reported increases in their satisfaction with their sex life, their sexual activity, their sex-seeking activity, and their use of protection for sex during COVID-19 pandemic, respectively.” – Similar comment as above.
- Similar changes as noted in the above two comments could be done for the first two sentences of the discussion section.
- Results – please provide a bit more information on how you got your results. Specifically, please shortly describe what variables you entered in which steps and so on.
- Your second paragraph of the discussion section needs more transition from the previous one.
- “The increased risk of contracting COVID-19 and sexually transmitted diseases should also be investigated in people who engage in sexual activity more but use protection less during the pandemic” – how many such people were in your sample?
- I am curious as to why relationship status wasn’t one of your factors. I think this variable, will have a significant impact, especially on sex-seeking behaviors. If someone is in a relationship, they don’t necessarily need to use apps and meet with strangers to engage in sex, as presumably they have a partner at home who is already sharing their germs.
- I would like the authors to estimate why their results are different from previous studies regarding gender. You noted that previous research found women more susceptible, but your findings demonstrate that men are the ones who had the change in their sexual activity. I think some explanation as to the inconsistency in the pattern is needed.
Author Response
Comment 1
“Considering the importance of sexuality to health, sexual counseling and risk reduction are aided by an understanding of how the pandemic has changed peoples’ sex life and what factors are associated with such change” – could be restructured to be clearer.
Response
Thank you for your comment. We revised this sentence into “Given the importance of sexuality to health, further study is needed to examine how the COVID-19 pandemic has changed peoples’ sex life and what factors are associated with such change [12].” Please refer to line 90-92.
Comment 2
“2. Factors Related to the Change in Sex Life” – There are likely multiple changes, so I think it would be better to rephrase the title as “Factors relating [or related] to the changes in sex life”
Response
Thank you for your comment. In the revised manuscript we divided the paragraph “Factors related to the Change in Sex Life” into two paragraphs titled “1.3. Associations of Risk Perception and General Anxiety with Changes in Sex Life” and “1.4. Demographic Factors of Changes in Sex Life” in Introduction section. Please refer to line 93 and 110.
Comment 3
I think additional clarity as to why you think a person’s sexual orientation would play a role in changes in sex during COVID-19 pandemic is necessary. You note: “In addition to gender and age, sexual orientation should also be analyzed as an influence on changes in sex life during epidemics. A study hypothesized that in times of a pandemic when people are particularly susceptible to mental health difficulties, messaging that frames sex as being dangerous has adverse psychological effects [7]. In this case, sexual minority individuals may be particularly vulnerable to sexual stigma, considering the historical trauma that sexual minorities have faced in other pandemics, such as that of AIDS [7].” In particular, the last sentence needs to be backed up further.
Response
Thank you for your comment. We revised the paragraph about the possible impacts of the COVID-19 pandemic on sex life of sexual minority individuals as below. Please refer line 118-134.
“A study hypothesized that in times of a pandemic when people are particularly susceptible to mental health difficulties, messaging that frames sex as being dangerous has adverse psychological effects [12]. There is no evidence that the COVID-19 can be transmitted via either vaginal or anal intercourse; however, there is evidence of oral-fecal transmission of the COVID-19 and that implies that anilingus may represent a risk for infection [30]. In this case, sexual stigma may have a negative role for sex life of sexual minority individuals [12]. From a historical perspective, human immunodeficiency virus and acquired immunodeficiency syndrome (AIDS) was conceived of as a "gay plague," by analogy with the sudden, devastating epidemics of the past [31]. Considering the historical trauma of the AIDS epidemic, sexual minority individuals may voluntarily or in voluntarily restrict their sex life. In addition to sexual stigma, sexual minority individuals may experience more severe impacts of the COVID-19 pandemic on their daily lives. A study on 1051 men who have sex with men (MSM) in the United States found that most of the participants had no change in condom access or use, many participants had adverse impacts of the COVID-19 pandemic to general wellbeing, social interactions, money, food, drug use and alcohol consumption; especially many reported fewer sex partners and COVID-19 related barriers to HIV and sexually transmitted diseases testing, pre-exposure prophylaxis access, and HIV treatment [32]. The authors also suggested that additional studies of COVID-19 epidemiology among sexual minority population are needed [32].”
Comment 4
“Facebook users were eligible for this study if they were aged ≥20 years and living in Taiwan.” – If they were at least 20 years of age, would fit better in this sentence.
Response
Thank you for your comment. We revised it into “if they were at least 20 years of age.” Please refer line 155.
Comment 5
“Scores of 0 or 1 indicated a decrease, a score of 2 indicated no change, and scores of 3 or 4 indicated an increase.” I believe this sentence is not necessary, as this information is incorporated in the previous sentence.
Response
Thank you for your comment. We deleted it from the revised manuscript. Please refer line 187.
Comment 6
“The Cronbach’s α for this section was 0.759” – for this measure, would fit better in this sentence.
Response
Thank you for your comment. We added “for this measure” into the sentences. Please refer line 202 and 210.
Comment 7
“As for changes in sex life, 13.4%, 13.5%, 6.7%, and 1.4% of respondents reported decreases in their satisfaction with their sex life, their sexual activity, their sex-seeking activity, and their use of protection for sex during the COVID-19 pandemic, respectively” could be improved by editing the sentence in the following way: “Respondents reported decreases in their satisfaction with their sex life (13.4%), their sexual activity (13.5%), their sex-seeking activity (6.7%), and their use of protection for sex (1.4%) during the COVID-19 pandemic” or something along these lines.
Response
Thank you for your comment. We edited this sentence based on your suggestion. Please refer line 233-235 in Results section and line 277-278 in Discussion section.
Comment 8
“By contrast, 1.9%, 2.9%, 1.9%, and 1.6% of respondents reported increases in their satisfaction with their sex life, their sexual activity, their sex-seeking activity, and their use of protection for sex during COVID-19 pandemic, respectively.” – Similar comment as above. Similar changes as noted in the above two comments could be done for the first two sentences of the discussion section.
Response
Thank you for your comment. We edited this sentence based on your suggestion. Please refer line 235-237 in Results section and line 278-280 in Discussion section.
Comment 9
Results – please provide a bit more information on how you got your results. Specifically, please shortly describe what variables you entered in which steps and so on.
Response
Thank you for your suggestion. We added more information on the steps of statistical analysis as below.
“Table 2 presents the multivariate logistic regression results for the associations of changes in each aspect of sex life (dependent variables) with sexual orientation, gender, age, risk perception of COVID-19, and general anxiety (independent variables).” Please refer line 243-245.
“The product interaction terms between gender, age, sexual orientation, risk perception, and general anxiety that were significantly associated with each aspect of change in sex life were selected into multiple logistic regression models to examine the moderators (Table 3).” Please refer line 258-260.
Comment 10
Your second paragraph of the discussion section needs more transition from the previous one.
Response
We added the sentence as below to serve as the transition from the first to the second paragraph of Discussion section. Please refer line 290-291.
“The present study found that risk perception, general anxiety gender, age, and sexual orientation related to various aspects of changes in sex life during the COVID-19 pandemic.”
Comment 11
“The increased risk of contracting COVID-19 and sexually transmitted diseases should also be investigated in people who engage in sexual activity more but use protection less during the pandemic” – how many such people were in your sample?
Response
We added the proportion of respondents who decreased use of protection for sex into the revised manuscript. Please refer line 286-287.
“Although only 1.4% of respondents reported the decreased use of protection for sex in the present study,…”
Comment 12
I am curious as to why relationship status wasn’t one of your factors. I think this variable, will have a significant impact, especially on sex-seeking behaviors. If someone is in a relationship, they don’t necessarily need to use apps and meet with strangers to engage in sex, as presumably they have a partner at home who is already sharing their germs.
Response
Thank you for your remining. We listed it as one of the limitations of this study as below. Please refer line 356-360.
“First, the present study did not ask participants about their current sexual relationship status or relationship quality, which are major factors affecting people’s sexual satisfaction during a pandemic like Covid-19 where people's social lives may have been curtailed. Sex between intimate couples can be an activity to support psychologically fragile people living in restricted areas for longer quarantine periods [54]”
Comment 13
I would like the authors to estimate why their results are different from previous studies regarding gender. You noted that previous research found women more susceptible, but your findings demonstrate that men are the ones who had the change in their sexual activity. I think some explanation as to the inconsistency in the pattern is needed.
Response
Thank you for your comment. We added more discussion about gender difference in changes of sexual activities as below. Please refer line 314-328.
“A study on Turkish women found that despite the increased frequency of sexual intercourse during the COVID-19 pandemic, quality of sexual life decreased during the pandemic [15]. The present study extends the scope of genders in participants and found that men were more likely to report a change, whether an increase or decrease, in their frequencies of sexual activity, sex-seeking activities, and use of protection for sex. A UK study also noted that men in the general population were more likely to engage in sexual activity during COVID-19 self-isolation [27]. Both biological and socioenvironmental factors may influence sexual behaviors across genders. Research on 3- to 6-year-old children found significant gender differences in sexual behavior [43]. Gender stereotypes in the societies may also affect the way people behave in sexual activities [44,45]. There might be multiple etiologies accounting for gender difference in changes of sexual activities during the COVID-19 pandemic and warrants further study. Moreover, research in China has demonstrated that women tend to be more psychologically affected by the COVID-19 outbreak with respect to stress, anxiety, depression, and posttraumatic stress symptoms [46,47]. The results of previous and present study indicated that the COVID-19 pandemic might have various impacts on sexual activities differently compared with and psychological wellbeing, as well as that changes in sex life might not be influenced by psychological status only.”